# Impact of Coenzyme Q10 Supplementation on Skeletal Muscle Respiration, Antioxidants, and the Muscle Proteome in Thoroughbred Horses

**DOI:** 10.3390/antiox12020263

**Published:** 2023-01-24

**Authors:** Marisa L. Henry, Lauren T. Wesolowski, Joe D. Pagan, Jessica L. Simons, Stephanie J. Valberg, Sarah H. White-Springer

**Affiliations:** 1Department of Large Animal Clinical Sciences, College of Veterinary Medicine, Michigan State University, East Lansing, MI 48824, USA; 2Department of Animal Science, College of Agriculture and Life Sciences, Texas A&M University and Texas A&M AgriLife Research, College Station, TX 77843, USA; 3Kentucky Equine Research, Versailles, KY 40383, USA

**Keywords:** antioxidants, mitochondria, high-resolution respirometry, reactive oxygen species, electron transport chain

## Abstract

Coenzyme Q10 (CoQ10) is an essential component of the mitochondrial electron transfer system and a potent antioxidant. The impact of CoQ10 supplementation on mitochondrial capacities and the muscle proteome is largely unknown. This study determined the effect of CoQ10 supplementation on muscle CoQ10 concentrations, antioxidant balance, the proteome, and mitochondrial respiratory capacities. In a randomized cross-over design, six Thoroughbred horses received 1600 mg/d CoQ10 or no supplement (control) for 30-d periods separated by a 60-d washout. Muscle samples were taken at the end of each period. Muscle CoQ10 and glutathione (GSH) concentrations were determined using mass spectrometry, antioxidant activities by fluorometry, mitochondrial enzyme activities and oxidative stress by colorimetry, and mitochondrial respiratory capacities by high-resolution respirometry. Data were analyzed using mixed linear models with period, supplementation, and period × supplementation as fixed effects and horse as a repeated effect. Proteomics was performed by tandem mass tag 11-plex analysis and permutation testing with FDR < 0.05. Concentrations of muscle CoQ10 (*p* = 0.07), GSH (*p* = 0.75), and malondialdehyde (*p* = 0.47), as well as activities of superoxide dismutase (*p* = 0.16) and catalase (*p* = 0.66), did not differ, whereas glutathione peroxidase activity (*p* = 0.003) was lower when horses received CoQ10 compared to no supplement. Intrinsic (relative to citrate synthase activity) electron transfer capacity with complex II (E_CII_) was greater, and the contribution of complex I to maximal electron transfer capacity (FCR_PCI_ and FCR_PCIG_) was lower when horses received CoQ10 with no impact of CoQ10 on mitochondrial volume density. Decreased expression of subunits in complexes I, III, and IV, as well as tricarboxylic acid cycle (TCA) enzymes, was noted in proteomics when horses received CoQ10. We conclude that with CoQ10 supplementation, decreased expression of TCA cycle enzymes that produce NADH and complex I subunits, which utilize NADH together with enhanced electron transfer capacity via complex II, supports an enhanced reliance on substrates supplying complex II during mitochondrial respiration.

## 1. Introduction

Coenzyme Q10 (CoQ10), 2,3-dimethoxy-5-methyl-6-decaprenyl-1,4-benzoquinone, is a potent fat-soluble antioxidant present in highest concentrations in the inner mitochondrial membrane [1]. There, CoQ10 acts as a redox carrier, transferring electrons from complexes I or II to complex III in the mitochondrial electron transfer system, eventually generating adenosine triphosphate (ATP) via oxidative phosphorylation [2]. Among its other functions, CoQ10 acts as a redox carrier, activator of uncoupling proteins, and modulator of the permeability transition pore [1].

A previous equine study found that muscle CoQ10 concentrations did not increase with 30 d of supplementation of 1600 mg of micellized ubiquinol combined with 10 g N-acetyl cysteine [3]. There were, however, significant changes in the muscle proteome that primarily involved mitochondrial tricarboxylic acid cycle (TCA) proteins. Whether this altered mitochondrial protein expression was the result of CoQ10 or N-acetyl cysteine supplementation could not be discerned from this study [3]. A supplementation trial using CoQ10 alone would be necessary to determine the sole effect of CoQ10 on the muscle proteome.

Although muscle CoQ10 concentrations did not increase with micellized ubiquinone supplementation [3], it is possible that CoQ10 enhances skeletal muscle mitochondrial respiratory function. A previous study of human patients with mitochondrial myopathies found that CoQ10 supplementation increased skeletal muscle mitochondrial function measured by phosphorous magnetic resonance spectroscopy, suggesting that the efficiency of respiration was increased independent of mitochondrial enzyme deficits [4]. To our knowledge, the effect of CoQ10 supplementation on mitochondrial oxidative phosphorylation has not been assessed using newer techniques such as high-resolution respirometry (HRR). Respirometry assesses the capacities of the mitochondrial electron transfer system by quantifying in situ oxygen consumption. Specifically, respirometry can separately quantify capacities of complexes I and II as well as measure capacities of electron transfer both coupled and uncoupled from ATP production [5,6]. HRR has recently been used to evaluate the effects of aging or dietary intervention in horses using saponin-permeabilized muscle fibers [5,6,7,8]. The application of HRR to muscle samples obtained from horses after chronic CoQ10 supplementation would be ideal to determine if CoQ10 supplementation impacts mitochondrial respiration.

The objective of this study was to determine the impacts of 30 d of 1600 mg CoQ10 supplementation on skeletal muscle CoQ10 concentrations, as well as muscle antioxidant balance, mitochondrial oxidative phosphorylation and electron transfer capacities, and the proteome. We hypothesized that supplementation with 1600 mg CoQ10 (micellized ubiquinol) for 30 d would increase mitochondrial oxidative phosphorylation capacities, increase antioxidants, and alter mitochondrial protein expression.

## 2. Materials and Methods

### 2.1. Horses

Six Thoroughbred horses (3 geldings and 3 mares, mean age ± SD 8.7 ± 2.2 y, body weight 542 ± 21 kg BW) were used for the present study. Horses were primarily housed in grass paddocks in groups of 3 based on sex for 19 to 20 h/d and were otherwise stalled in 3.7 × 3.7 m bedded stalls prior to and following exercise. Horses had ad libitum access to predominately Bahiagrass pasture and water while in paddocks and ad libitum access to timothy hay and water while stalled (Table 1). In addition, horses received 4 to 5 kg/d as-fed (3.6 to 4.5 kg/d DM) of OBS Sport feed (Ocala Breeders Supply, Ocala, FL, USA) BID to maintain a body condition score (BCS) between 5 and 6 [9]. The basal diet was formulated to meet or slightly exceed dietary requirements for horses in light work [10].

Horses exercised 6 d/wk with an alternating schedule of treadmill and under saddle exercise. Three d/wk, horses were exercised on a high-speed treadmill for 20 min/d at a 3 to 5° incline at a walk, trot, canter (6 to 7 m/s), and gallop (8 to 10 m/s). The remaining 3 d/wk, horses walked for 30 min on a 6-horse panel mechanical walker measuring approximately 20.1 m in diameter (EquiGym, LLC, Lexington, KY, USA) followed by 30 min under saddle exercise at the walk, trot, and canter. Horses were accustomed to this exercise program 7 to 8 wk prior to initiation of the study.

### 2.2. Study Design

A randomized cross-over study design was utilized with two 30-d periods separated by a 60-d washout. During the first period, horses were randomly assigned to receive either (1) supplementation with 1600 mg CoQ10/d through 15 mL of Nano-Q10 (Kentucky Equine Research, Versailles KY) divided into two doses and administered orally with a dosing syringe; or (2) a control treatment in which horses received no additional supplementation. Following the washout period, horses switched treatment groups for the second 30-d supplementation period. During the washout period, all horses received only the control diet and continued the same weekly exercise protocol.

### 2.3. Muscle Samples

Gluteus medius muscle samples were collected at the end of each period prior to daily CoQ10 supplementation and exercise [11]. Horses were sedated with intravenous administration of detomidine hydrochloride (Dormosedan; Zoetis Services LLC, Parsippany, NJ, USA) at doses recommended by the manufacturer. The collection site was clipped free of hair and surgically cleaned with povidone scrub and 70% ethyl alcohol. Between 0.1 and 0.5 mL 2% Lidocaine (VetOne, MWI Veterinary Supply Company, Boise, ID, USA) was administered intradermally to desensitize the area. A 14-gauge needle was used to puncture the skin followed by insertion of a 14-gauge tissue collection needle (SuperCore Semi-Automatic Biopsy Instrument, Argon Medical Devices, Frisco, TX, USA) to a fixed depth of 5 cm to collect muscle tissue. Tissue was collected from the left side of the horse at the end of period 1 and the right side of the horse at the end of period 2. Samples were aliquoted and either flash frozen in liquid nitrogen and stored at −80 °C until future enzyme activity analyses or placed in mitochondrial preservation solution (BIOPS; 10 mM Ca-EGTA buffer, 0.1 µM free calcium, 20 mM imidazole, 20 mM taurine, 50 mM K-MES, 0.5 mM dithiothreitol, 6.56 mM MgCl_2_, 5.77 mM ATP, and 15 mM phosphocreatine; pH 7.1) and stored on ice or at 4 °C until high-resolution respirometry was performed. Upon finishing collection, the site was cleaned with 70% ethyl alcohol and sealed using aluminum bandage (AluShield Spray, MWI Veterinary Supply Company, Boise, ID, USA).

### 2.4. CoQ10 Analysis

CoQ10 analysis of muscle samples was performed at the Michigan State University Mass Spectrometry and Metabolomics Core using a high-resolution/accurate-mass (HR/AM) UHPLC-MS/MS system consisting of a Thermo Vanquish UHPLC interfaced with Thermo Q-Exactive according to Pandey 2018 [12]. Data were analyzed as described [3]. Briefly, approximately 10 mg of tissue was homogenized in 500 mL of 95:5 ethanol:2-propanol solution containing 500 ng/mL CoQ4 internal standard with 125 mg of butylated hydroxytoluene pre-dried in the homogenization tube with a bead homogenizer (bullet blender, Next Advance, Troy, NY, USA). CoQ10 was extracted with a 2:1 solution of hexanes to water, then the organic layer was removed and evaporated. Within 24 h of quantification, samples were reconstituted in 2 mL of ethanol containing 0.3 M hydrochloric acid. Data were processed using Xcalibur software version 4.1.31.9 and accounted for different final dilution volumes.

### 2.5. High-Resolution Respirometry

Muscle samples collected into ice-cold BIOPS and stored on ice or at 4 °C were analyzed for mitochondrial oxidative phosphorylation (P) and electron transfer system capacities (E) using high-resolution respirometry within 24 h of collection. Immediately prior to analysis, samples were saponin permeabilized as described previously [5]. Permeabilized fibers were then rinsed in mitochondrial respiration solution (MiR05; 110 mM sucrose, 60 mM potassium lactobionate, 0.5 mM EGTA, 3 mM MgCl_2_∙6H_2_O, 20 mM taurine, 10 mM KH_2_PO_4_, 20 mM HEPES, 1 g/L BSA, pH 7.1) for 10 min at 4 °C. Approximately 1.5–2.5 mg (wet weight) of rinsed fibers were then immediately added to each chamber of an Oroboros Oxygraph-2k (O2k; Oroboros, Innsbruck, Austria) containing MiR06 (MiR05 + 280 U/mL catalase) and 20 mM creatine. Chambers were maintained at 37 °C and in hyperoxic conditions (200 to 650 µM O_2_) through the addition of 200 mM H_2_O_2_. The previously described [13] substrate uncoupler inhibitor titration protocol for this study was as follows: (1) complex I substrates, pyruvate (5 mM), and malate (1 mM), to determine mitochondrial proton leak (LEAK); (2) adenosine diphosphate (ADP; 2.5 mM), to quantify complex I-supported P (P_CI_); (3) glutamate (10 mM), an additional complex I substrate (P_CIG_); (4) cytochrome *c* (cyt *c*, 10 µM), to measure integrity of the outer mitochondrial membrane; (5) the complex II substrate, succinate (10 mM), to measure maximal P (P_CI+II_); (6) uncoupler carbonyl cyanide 3-chlorophenylhydrazone (CCCP, 0.5 µM steps), to attain maximal noncoupled E (E_CI+II_); (7) a complex I inhibitor, rotenone (0.5 µM), to measure complex II-supported E (E_CII_); and (8) a complex III inhibitor, antimycin A (2.5 µM), to quantify non-mitochondrial residual O_2_ consumption. All data were normalized to residual O_2_ consumption. Respiration data are presented either relative to tissue weight (integrative), CS activity (mitochondrial volume density; intrinsic), or as a ratio of contribution to maximal electron transfer capacity (flux control ratio, FCR).

### 2.6. Mitochondrial Enzyme Activities

Citrate synthase (CS) and cytochrome *c* oxidase (CCO) activities were determined as measures of mitochondrial volume density and function, respectively, using kinetic colorimetry [5,14]. Previously cryopulverized (Spectrum™ Bessman Tissue Pulverizer; Spectrum Laboratories, Inc., Rancho Dominguez, CA, USA) muscle powder was sonicated in sucrose homogenization buffer (20 mM Tris, 40 mM KCl, 2 mM EGTA, 250 mM sucrose) with 1 part 5% detergent (n-Dodecyl β-D-maltoside; Sigma D4641) 3 times for 3 s each while on ice. Samples were centrifuged at 11,000× *g* for 3 min at 0 °C, and the supernatant homogenate was aliquoted and stored at −80 °C until analysis. Enzyme activities were measured using a microplate reader (Synergy H1, BioTek Instruments, Winooski, VT, USA). Briefly, CS activity was determined by measuring the initial rate of reaction of free CoA-SH with DTNB at 412 nm at 37 °C, and CCO activity was determined by measuring the linear rate of oxidation of fully reduced cytochrome *c* at 550 nm at 37 °C. Both assays utilized 80-fold diluted muscle homogenate and were analyzed in duplicate. The intra-assay and inter-assay coefficient of variation (CV) for CS activity was 2.3% and 2.8%, respectively. CCO activity was measured on a single plate with an intra-assay CV of 2.4%. Enzyme activities were normalized to total protein content, which was determined using the Coomassie Bradford Protein Assay Kit (Thermo Fisher Scientific, Waltham, MA, USA). CCO activity was also normalized to CS activity (intrinsic) as a measure of function per mitochondria within the sample [15].

### 2.7. Antioxidant Analyses

Glutathione (GSH) was measured using a high-performance liquid chromatography mass spectrometry analysis as previously described [3]. Briefly, approximately 30 mg of tissue was homogenized in a bead homogenizer with 500 uL 1× radioimmunoprecipitation assay (RIPA) buffer, reduced with Tris(2-carboxyethyl)phosphine hydrochloride and N-ethylmaleimide. An internal standard of 20 uM GSH ammonium salt D-5 (Toronto Research Chemicals, Toronto, ON, Canada) was utilized to standardize all samples and standards. Chromatographic separation was performed using a Phenomenex Kinetex 1.7 um EVO C18 100A (50 mm × 2.1 mm) (Phenomenex, Torrance, CA, USA) column.

Muscle glutathione peroxidase (GPx), superoxide dismutase (SOD), and catalase (Cat) activities were measured using colorimetric assay kits (Cayman Chemical, Ann Arbor, MI, USA) according to the manufacturer’s instructions. For the SOD assay, 10 mg of tissue was homogenized in a bead homogenizer using 500 uL buffer at pH 7.2 buffer containing 20 mM 2-[4-(2-hydroxyethyl)piperazine-1-yl]ethanesulfonic acid, 1 mM ethylene glucol-bis(b-aminoethyl ether)-N,N,N′,N′-tetraacetic acid, 210 mM mannitol, and 70 mM sucrose and centrifuged at 1500× *g* for 5 min at 4 °C. For the GPx assay, 10 mg of tissue from was homogenized in 500 µL pH 7.5 buffer containing 50 mM Tris-HCl, 5 mM ethylenediaminetetraacetic acid, and 1 mM dithiothreitol and centrifuged at 10,000× *g* for 15 min at 4 °C. For the Cat assay, 10 mg of tissue was homogenized in 500 uL pH 7.0 buffer containing 50 mM potassium phosphate and 1 mM ethylenediaminetetraacetic acid and centrifuged at 10,000× *g* for 15 min at 4 °C. All supernatants were collected immediately and analyzed in triplicate. GPx, SOD, and Cat activities were measured on a single plate with an intra-assay CV of 2.9%, 2.6%, and 4.7% respectively.

The standard BCA assay kit (Catalog #23225; Thermo Scientific) was used to determine the protein concentrations for CoQ10 and GSH. The reducing agent BCA assay kit (Catalog #23250; Thermo Scientific) was used to determine the protein content for Cat, GPx, and SOD homogenates.

### 2.8. Malondialdehyde (MDA) Analysis

Using a commercially available kit (Northwest Life Science Specialties, Vancouver, WA, USA), muscle malondialdehyde (MDA) concentration, a marker of lipid peroxidation, was measured as previously described [16]. Briefly, cryopulverized muscle powder was diluted 1 mg tissue (wet weight) to 10 μL of assay buffer then sonicated while on ice and centrifuged at 11,000× *g* for 10 min at 0 °C. The supernatants were collected and stored at −80 °C until analysis. Samples were analyzed in triplicate with intra- and inter-assay CV of 2.2% and 1.6%, respectively. Malondialdehyde concentration was normalized to total protein quantified by the Coomassie Bradford Protein Assay kit (Thermo Fisher Scientific).

### 2.9. Proteomics

Proteomic analysis was performed on muscle samples from 5 horses in both treatment groups using an 11-plex plates for tandem mass tag MS/MS quantification analysis at the Michigan State University Proteomics Core. Horses were selected based on the respirometry data. Protein was extracted from muscle samples using a radioimmunoprecipitation lysis buffer and protease inhibitor and pelleted prior to submission. Protein concentrations were determined by standard BCA assay. From each sample, 500 mg of protein was digested in trypsin with a Filter-Aided Sample preparation protocol and spin ultrafiltration unit cutoff of 30,000 Da [17]. Reverse-phase C18 SepPaks were used to de-salt the resulting peptides (Waters Corporation, Milford, MA, USA) which were then dried by vacuum centrifugation. Peptide quantification was verified by colorimetric peptide concentration using 5 mL from each sample digest. Isobaric labeling, gel fractionation, and LC/MS/MS analysis were performed.

Proteomic analysis was performed using Scaffold Q+ version 5.0.1 (Proteome Software Inc., Portland, OR, USA). Peptide identifications were accepted if they could be established at greater than 95.0% probability by the Scaffold Local FDR algorithm. Protein identifications were accepted if they could be established at greater than 99.9% probability and contained at least 2 identified peptides. Protein probabilities were assigned by the Protein Prophet algorithm [18]. Proteins that contained similar peptides and could not be differentiated based on MS/MS analysis alone were grouped to satisfy the principles of parsimony. Proteins sharing significant peptide evidence were grouped into clusters. Channels were corrected by the matrix in all samples according to the algorithm described in i-Tracker [19]. Normalization was performed iteratively (across samples and spectra) on intensities, as described in Statistical Analysis of Relative Labeled Mass Spectrum data from Complex Samples Using ANOVA [20]. Medians were used for averaging. Spectra data were log-transformed, pruned of those matched to multiple proteins, and weighted by an adaptive intensity weighting algorithm. Differentially expressed proteins were determined by applying Permutation Test with an adjusted *p* value of *p* < 0.0027 corrected by Benjamini-Hochberg. Significant proteins were grouped according to their cellular functions.

### 2.10. Statistical Analysis

Biochemical and HRR data were analyzed using PROC MIXED in SAS (version 9.4, SAS Institute, Inc., Cary, NC) using a mixed model equation with period (period 1 or period 2), supplementation (CoQ10 or control), and period × supplementation as fixed effects and horse as a repeated effect. Significance was set at *p* ≤ 0.05, and data were presented as means ± SD. Graphs were created using GraphPad Prism software version 9.3.1 (GraphPad Software San Diego, CA, USA).

## 3. Results

### 3.1. Skeletal Muscle CoQ10 Concentrations

Although there was no significant difference in muscle CoQ10 concentrations between CoQ10 treatment and the control diet, there was a trend toward lower concentrations of CoQ10 with supplementation (*p* = 0.07) (Figure 1).

### 3.2. Mitochondrial Enzyme Activities Capacities and High Resolution Respirometry

CS activity, integrative and intrinsic CCO activities (Figure 2), and integrative (per mg tissue) mitochondrial capacities (Appendix A) were unaffected by treatment, period, or the treatment × period interaction.

Intrinsic (per CS activity) LEAK and oxidative phosphorylation capacities with complex I (P_CI_ and P_CIG_) were unaffected by period, treatment, or the treatment × period interaction (Figure 3A–C). Conversely, intrinsic maximal coupled oxidative phosphorylation capacity (P_CI+II;_
*p* = 0.07) and maximal noncoupled electron transfer capacity (E_CI+II;_
*p* = 0.08) tended to be greater when horses received CoQ10 compared to the control diet with no period or treatment × period interaction (Figure 3D,E). Finally, intrinsic electron transfer capacity with complex II only (E_CII_) was impacted by the interaction of treatment and period (*p* = 0.05); E_CII_ was greater in CoQ10 compared to control horses following period 1 (*p* = 0.005) but was not different among horses after period 2 (Figure 3F).

The contribution of LEAK to maximal electron transfer capacity (flux control ratio; FCR_LEAK_) was not impacted by CoQ10 supplementation or period. FCR_PCI_ and FCR_PCIG_ were lower with CoQ10 supplementation compared to the control diet (*p* = 0.03) but were unaffected by period and the treatment × period interaction (Figure 4A–C). A treatment *×* period interaction was noted, with FCR_PCI+II_ being greater in horses receiving CoQ10 following period 2 compared to period 1 (*p* = 0.05) but no difference in FCR_PCI+II_ in horses receiving the control diet between periods (Figure 4D). FCR_ECII_ was unaffected by period or treatment *×* period interaction (Figure 4E).

### 3.3. Antioxidants

GPx activity was lower with CoQ10 supplementation compared to the control diet (*p* = 0.003) with a supplement × period interaction (*p* = 0.02) (Figure 5A). SOD and Cat activities and GSH and MDA concentrations were unaffected by treatment, period, and the interaction of treatment and period (Figure 5B–E).

### 3.4. Proteomics

In total, 834 total unique proteins were identified. Of these, 38 were differentially expressed, 8 with increased expression and 30 with decreased expression comparing CoQ10 supplementation with the control diet (Table 2). All proteins found within mitochondria were downregulated with CoQ10 supplementation, including two subunits of complex I (out of 45 mammalian subunits), two of three subunits of complex III, and four subunits of ATP synthase (out of 29 subunits). Other differentially expressed mitochondrial proteins included downregulation of four enzymes in the TCA cycle, five enzymes in fatty acid metabolism, two voltage-dependent ion channels, one chaperone protein, and a membrane channel for GSH import into the mitochondria (Table 2, Figure 6). The downregulated TCA cycle proteins included malate dehydrogenase (MDH2), aconitate hydrase (ACO2), isocitrate dehydrogenase 2 (IDH2), and 2-oxoglutarate dehydrogenase (OGDH). Within the inner mitochondrial membrane, an additional electron transfer protein, NAD(P) transhydrogenase (NNT), which generates nicotinamide adenine dinucleotide phosphate (NADPH), was downregulated. In addition, seven proteins in the sarcomere and sarcoplasmic reticulum (four upregulated and three downregulated), four in glycolysis and gluconeogenesis (three upregulated and one downregulated), and two miscellaneous proteins were differentially expressed (Table 2).

## 4. Discussion

The present study assessed the impact of 30 d of supplementation with 1600 mg CoQ10 on muscle CoQ10 concentrations, mitochondrial oxidative phosphorylation, the muscle proteome, and muscle antioxidant status. We found that maximal intrinsic oxidative phosphorylation and electron transfer capacities tended to be greater with CoQ10 supplementation, which was accompanied by a lower contribution of complex I to maximal electron transfer capacity. Proteomic data revealed small but significantly decreased expression of mitochondrial proteins within complexes I, III, and IV, as well as proteins involved in shuttling GSH into the mitochondria. Muscle CoQ10 concentrations trended lower and GPx activity was decreased with CoQ10 supplementation but there was no indication of oxidative stress assessed by MDA. Thus, chronic CoQ10 supplementation appears to impact mitochondrial respiration and antioxidant status, favoring substrates supplying FADH_2_ for complex II rather than NADH for complex I.

In humans, after ingestion, CoQ10 is rapidly taken up by the liver, and a limited amount is released into the blood bound to circulating lipoproteins [21]. Previous studies have found that 14 d of CoQ10 supplementation in humans (100 mg fast-melt CoQ10 BID) significantly increased plasma CoQ10 concentrations in the CoQ10-supplemented groups compared to the control group [22]. In the present study, muscle CoQ10 concentrations did not increase with supplementation, in agreement with our previous study of CoQ10 and N-acetylcysteine supplementation of Thoroughbred racehorses for 30 d [3]. This is in contrast with reports of greater CoQ10 concentrations in muscle samples following 10 and 21 d of supplementation of 1 g of ubiquinol (Recovery Support, Anlon Nutrition, Kilcullen, Ireland) in fit Thoroughbreds [23]. The Thueson et al. [24] study, however, utilized different methodology for measuring CoQ10 and had concentrations that were 100× lower than those measured in our study. In agreement with our study, Cooke et al. reported no difference in human muscle CoQ10 concentrations after 14 d of CoQ10 supplementation and wide inter-individual variation in CoQ10 concentrations [22]. It is possible that muscle CoQ10 concentrations are saturated when levels are adequate and therefore no additional CoQ10 is taken up into muscle fibers. CoQ10 is highly lipophilic and has a high molecular weight and poor aqueous solubility, which could limit its absorption and muscle uptake [24].

One of the primary goals of this study was to determine if CoQ10 supplementation impacted mitochondrial respiration. We found no differences in CS activity whether horses received CoQ10 supplementation or the unsupplemented control diet, indicating that mitochondrial volume density was not impacted by CoQ10 supplementation. Integrative (relative to mg of protein) maximal oxidative phosphorylation and electron transfer capacities were also unaffected by CoQ10 supplementation. However, intrinsic (relative to CS activity) maximal oxidative phosphorylation and electron transfer capacities differed. When horses were supplemented with CoQ10, maximal oxidative phosphorylation and electron transfer capacities increased but intrinsic complex I-supported oxidative phosphorylation capacities (P_CI_ and P_CIG_) did not change. The increase in maximal capacities, therefore, appears to be due to an increase in complex II capacity. This is supported by the decrease in the contribution of complex I (FCR_PCI_ and FCR_PCIG_) to maximal electron transfer noted in horses receiving CoQ10 compared to the control diet. These data suggest that CoQ10 supplementation results in a preferential utilization of complex II over complex I for energy production.

An impact of CoQ10 supplementation on skeletal muscle was further suggested by the significant differences in protein expression identified within the muscle proteome and by differences in muscle GPx activity. The expressions of two subunits of complex I, two subunits of complex III, and four subunits of complex V were significantly decreased, as well as the expressions of TCA cycle proteins and enzymes involved in fat metabolism (Table 2 and Figure 6). Each complex in the electron transfer system contains multiple subunits; however, interestingly, the downregulated subunits with CoQ10 supplementation were of utmost importance to electron transport. In complex I, the subunits with decreased expression are those that directly transfer electrons from complex I to CoQ10 [25], and in complex III, the downregulated Rieske subunit directly transfers electrons from complex III to cytochrome c [26] (Figure 6). The fold change of the expression of the subunits is relatively small but their statistical significance and roles indicate that CoQ10 supplementation impacted the muscle proteome and, potentially, electron transfer.

In addition to linking electron transfer from complexes I and II to complex III, CoQ10 can also form a functional structure referred to as the CoQ10-junction [27]. This junction primarily facilitates the reduction of sulfur-based molecules that donate electrons directly to CoQ10, where they can be transferred to complex III provided there is ongoing FADH_2_ mediated electron transport through complex II [27]. FADH_2_-mediated electron transport flux is enhanced by increased fatty acid utilization. One reason CoQ10 supplementation could decrease the movement of electrons through complex I could be that the additional provision of CoQ10 enhances electron transfer via the CoQ10 junction and potentially enhances fatty acid utilization. In addition, the proteome identified decreased expression of MDH2, OGDH, and ACO2, which, within the TCA cycle, generate NADH, the substrate for complex I. Reduced TCA cycle activity could reduce the flow of electrons to complex I. Thus, our results suggest that CoQ10 supplementation in horses has the potential to impact the contributions of complex I and complex II to electron transport.

Complex I has been implicated as the primary generator of reactive oxygen species (ROS) in resting muscle compared to the other complexes of the electron transfer system [28]. Decreased electron flux through complex I might lead to decreased production of ROS. In humans, CoQ10 supplementation suppressed hydrogen peroxide (H_2_O_2_) levels during leak respiration, which is the state where mitochondrial ROS production is greatest [29]. In the present study, we did not identify evidence of oxidative stress based on the measurement of MDA concentrations. It is important to recognize, however, that muscle samples were taken at rest and that daily exercise was submaximal. Future research is warranted to determine if CoQ10 supplementation impacts markers of oxidative stress immediately after exercise in horses. This would be especially interesting considering the possibility that CoQ10 supplementation seems to favor electron transfer via complex II over complex I, which could lower the production of ROS.

Both CoQ10 and GSH are potent nonenzymatic antioxidants. In our study, we assessed total GSH and did not determine alteration in the ratio of GSSG/GSH due to CoQ10 supplementation, a more sensitive indicator of redox state. CoQ10 supplementation of horses in the present study resulted in a significant decrease in activity of an enzymatic antioxidant GPx, which serves to reduce oxidized proteins either on its own or using GSH as a co-factor. Notably, the proteomic data identified decreased expression of proteins involved in shuttling GSH into mitochondria. Expression of the SLC25A protein channel, which exports 2-oxoglutarate (2-OG) from the mitochondria and imports malate and GSH into the mitochondria, was decreased (Figure 5). In addition, IDH2, which produces 2-OG, and AST, which can convert oxaloacetate to 2-OG, had decreased expression, suggesting a decreased need for shuttling GSH into mitochondria. A decreased need to import GSH into mitochondria could arise from the potent antioxidant capacity of supplemented CoQ10.

The magnitude of significant differences arising from CoQ10 supplementation noted in our study was relatively small, and thus the supplement may have a limited effect on healthy horses performing a short period of submaximal exercise daily. It is possible that CoQ10 would have a greater impact on horses generating significantly more ROS such as endurance horses, maximally exercising racehorses, or horses with myopathies such as myofibrillar myopathy [30]. This potential impact remains to be studied.

## 5. Conclusions

Chronic CoQ10 supplementation in fit Thoroughbred horses did not increase muscle CoQ10 concentrations but did alter mitochondrial oxidative phosphorylation capacities, favoring the utilization of complex II over complex I for the movement of electrons during oxidative phosphorylation and electron transfer. We found decreased expression of subunits of complex I, III, and V and TCA proteins that produce the NADH that supplies electrons to complex I with CoQ10 supplementation. A potential compensatory decrease in GPx activity was found in horses receiving the potent CoQ10 antioxidant with concomitant downregulation of proteins involved in shuttling GSH into the mitochondria. Taken together, CoQ10 in equine muscle appears to function both as an antioxidant and to alter electron transfer through complex II over complex I without increasing mitochondrial volume density. The magnitude of changes measured in healthy horses supplemented with CoQ10 were relatively small, and their impact on performance was not resolved in the present study.

## Figures and Tables

**Figure 1 antioxidants-12-00263-f001:**
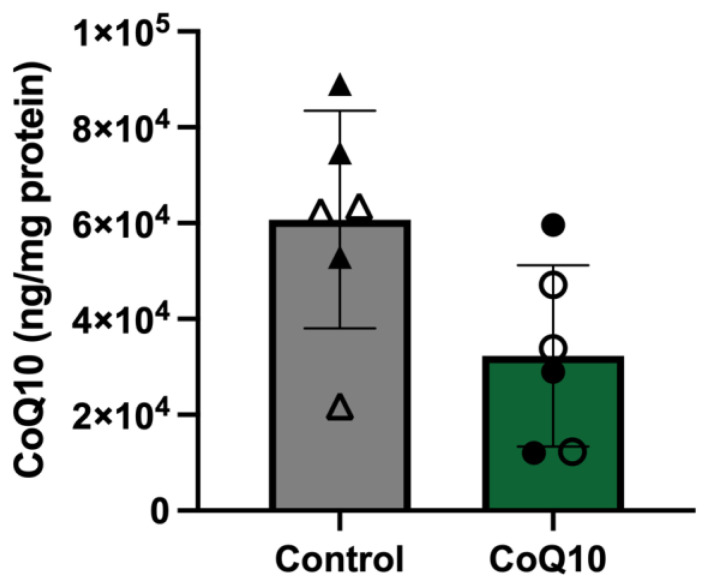
Skeletal muscle CoQ10 concentrations did not differ significantly (*p* = 0.07) between control and CoQ10 treatments. Open circles represent horses on the CoQ10 supplement during the first supplementation period, and closed circles represent horses on the CoQ10 supplement during the second supplementation period. Open triangles represent horses on the control diet during the second supplementation period, and closed triangles represent horses on the control diet during the first supplementation period.

**Figure 2 antioxidants-12-00263-f002:**
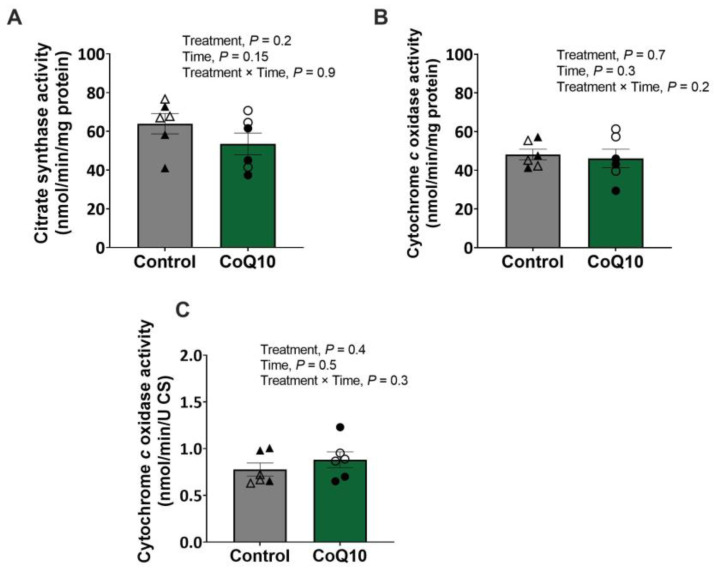
(**A**) CS activity and (**B**) integrative and (**C**) intrinsic CCO activities in the gluteus medius of fit Thoroughbred horses before and after 30 d supplementation of CoQ10 or control diet. Open circles represent horses on the CoQ10 supplement during the first supplementation period, and closed circles represent horses on the CoQ10 supplement during the second supplementation period. Open triangles represent horses on the control diet during the second supplementation period, and closed triangles represent horses on the control diet during the first supplementation period.

**Figure 3 antioxidants-12-00263-f003:**
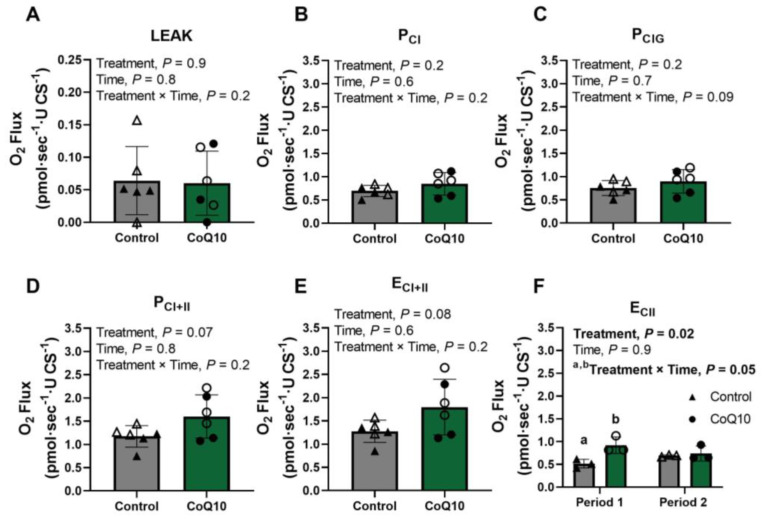
(**A**–**F**) Intrinsic (relative to CS activity) mitochondrial capacities in the gluteus medius of fit Thoroughbred horses before and after 30 d supplementation of CoQ10 or control diet. Open circles represent horses on the CoQ10 supplement during the first supplementation period, and closed circles represent horses on the CoQ10 supplement during the second supplementation period. Open triangles represent horses on the control diet during the second supplementation period, and closed triangles represent horses on the control diet during the first supplementation period. Within periods, different letters indicate significant treatment differences.

**Figure 4 antioxidants-12-00263-f004:**
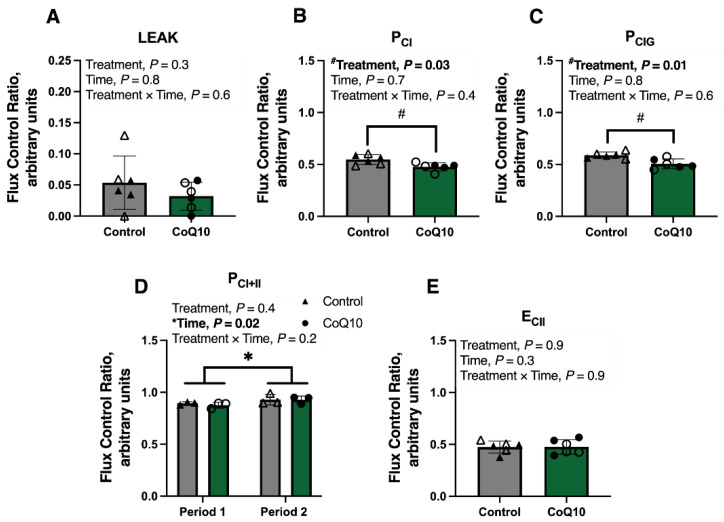
(**A**–**E**) Flux control ratios (FCR) of gluteus medius samples from fit Thoroughbred horses before and after 30 d supplementation of CoQ10 or control diet. Open circles represent horses on the CoQ10 supplement during the first supplementation period, and closed circles represent horses on the CoQ10 supplement during the second supplementation period. Open triangles represent horses on the control diet during the second supplementation period, and closed triangles represent horses on the control diet during the first supplementation period. # Regardless of time (period), control differs from CoQ10 (*p* ≤ 0.05). * Across treatments, period 1 differs from period 2 (*p* ≤ 0.05).

**Figure 5 antioxidants-12-00263-f005:**
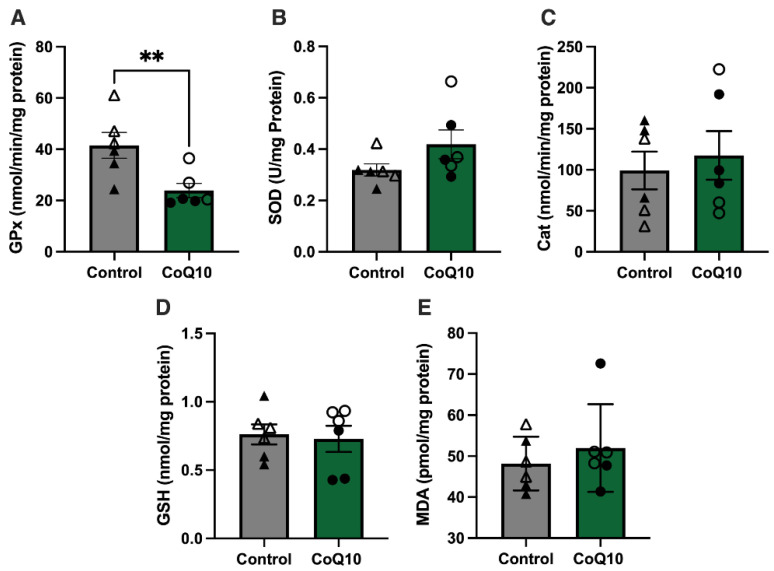
Skeletal muscle antioxidant activities and MDA concentrations relative to protein concentrations. (**A**) Activity of GPx in control and CoQ10 treatment groups. GPx activity was significantly lower in the CoQ10 supplemented horses (*p* = 0.003). (**B**) Activities of SOD in control and CoQ10 treatment groups were not significantly different with treatment (*p* = 0.17). (**C**) Activities of Cat in control and CoQ10 treatment groups were not significantly different with treatment (*p* = 0.66). (**D**) GSH concentrations in control and CoQ10 treatment groups were not significantly different (*p* = 0.75). (**E**) MDA concentrations in control and CoQ10 treatment groups were not significantly different (*p* = 0.47). Open circles represent horses on the CoQ10 supplement during the first supplementation period, and closed circles represent horses on the CoQ10 supplement during the second supplementation period. Open triangles represent horses on the control diet during the second supplementation period, and closed triangles represent horses on the control diet during the first supplementation period. ** *p* < 0.01.

**Figure 6 antioxidants-12-00263-f006:**
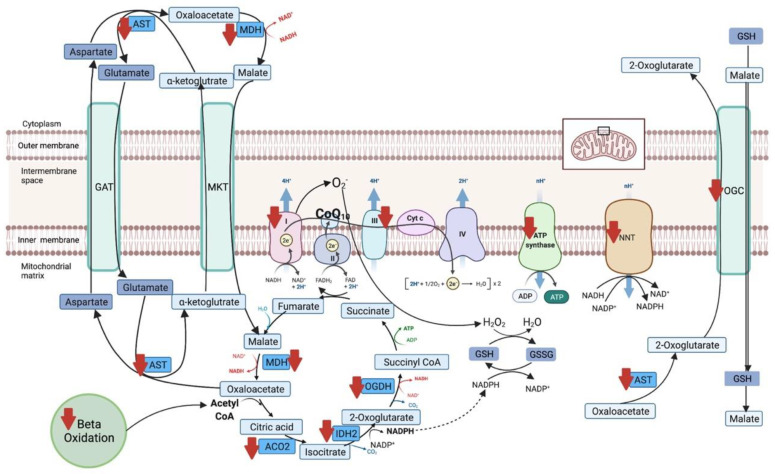
Proteins with significantly decreased expression (red arrow) in the mitochondria of horses on CoQ10 compared to the control. Horses with CoQ10 supplementation had a downregulation of 2 subunits of complex I, 2 subunits of complex III, and 4 subunits of ATP synthase, NAD(P) transhydrogenase (NNT), malate dehydrogenase (MDH), 2-oxoglutarate dehydrogenase (OGDH), NADP dependent isocitrate dehydrogenase (IDH2), aconitase (ACO2), aspartate aminotransferase (AST), and the 2-oxoglutarate transport channel (OGC). Created with BioRender.com, accessed on 2 December 2022.

**Table 1 antioxidants-12-00263-t001:** Nutrient composition of OBS sport feed, timothy hay, and Bahiagrass pasture offered to horses. All nutrients are presented on a 100% dry matter (DM) basis.

Nutrient	OBS Sport	Timothy Hay	Grass Pasture
DE, Mcal/kg	3.47	2.08	1.97
CP, %	15.95	12.75	18.05
ADF, %	24.5	34.9	34.8
NDF, %	37.85	59.75	59.90
Starch, %	4.6	0.9	2.15
Crude Fat, %	8.89	2.87	3.88
Ca, %	1.19	0.38	0.59
P, %	0.81	0.22	0.42
Mg, %	0.33	0.23	0.28
K, %	1.60	1.83	2.62
Na, %	0.33	0.02	0.03
Fe, ppm	530.5	148.0	276.5
Zn, ppm	181.5	28.0	47.5
Cu, ppm	68.5	7.0	66.0

**Table 2 antioxidants-12-00263-t002:** The gene identification for protein, protein name, *p* value adjusted for multiple test corrections, and log_2_ fold change for differential expressed proteins comparing CoQ10 supplementation to the control diet.

Gene ID	Protein Name	Adjusted *p* Value	Log_2_ Fold Change
	**Mitochondria**		
NDUFS8	NADH dehydrogenase [ubiquinone] iron-sulfur protein 8, mitochondrial	0.002	−0.12
NDUFB5	NADH dehydrogenase [ubiquinone] 1 beta subcomplex subunit 5, mitochondrial	0.001	−0.11
UQCRFS1	cytochrome b-c1 complex subunit Rieske, mitochondrial	0.002	−0.09
UQCRH	cytochrome b-c1 complex subunit 6, mitochondrial isoform X2	0.001	−0.14
ATP5F1c	ATP synthase subunit gamma, mitochondrial isoform X3	0.00091	−0.11
ATP5ME	ATP synthase subunit e, mitochondrial	0.0007	−0.14
ATP5PD	ATP synthase subunit d, mitochondrial isoform X2	0.00029	−0.11
MT-ATP6	ATP synthase subunit alpha, mitochondrial	<0.0001	−0.09
PHB2	prohibitin-2	0.001	−0.07
OGDH	Cluster of 2-oxoglutarate dehydrogenase, mitochondrial isoform X3	0.00097	−0.08
IDH2	Cluster of isocitrate dehydrogenase [NADP], mitochondrial	0.001	−0.07
ACO2	aconitate hydratase, mitochondrial	<0.0001	−0.09
MDH2	malate dehydrogenase, mitochondrial	<0.0001	−0.09
NNT	NAD(P) transhydrogenase, mitochondrial isoform X1	<0.0001	−0.06
HADHA	trifunctional enzyme subunit alpha, mitochondrial	0.00046	−0.06
ACAA2	3-ketoacyl-CoA thiolase, mitochondrial	0.002	−0.03
ETFB	electron transfer flavoprotein subunit alpha, mitochondrial	0.001	−0.12
ACADVL	very long-chain specific acyl-CoA dehydrogenase, mitochondrial isoform X6	<0.0001	−0.05
CRAT	carnitine O-acetyltransferase isoform X1	<0.0001	−0.06
AST	aspartate aminotransferase, mitochondrial	0.00044	−0.08
Aifm1	apoptosis-inducing factor 1, mitochondrial isoform X1	0.00098	−0.1
DLD	dihydrolipoyl dehydrogenase, mitochondrial	0.00038	−0.07
SLC25A	phosphate carrier protein, mitochondrial isoform X1	0.001	−0.09
VDAC1	voltage-dependent anion-selective channel protein 1	0.00082	−0.07
VDAC2	voltage-dependent anion-selective channel protein 2	0.001	−0.09
	**Sarcomere and SR**		
MYBPC2	myosin-binding protein C, fast-type	<0.0001	0.09
MYL1	myosin light chain 3	0.00012	−0.17
MYL2	myosin regulatory light chain 2, ventricular/cardiac muscle isoform	0.00037	−0.12
MYLK2	myosin light chain kinase 2, skeletal/cardiac muscle	0.00058	0.12
MYOM1	myomesin-1 isoform X4	0.00065	0.04
TNNT1	troponin I, slow skeletal muscle	0.001	−0.15
RYR1	Cluster of ryanodine receptor 1 isoform X1	0.00078	0.04
	**Glycolysis/gluconeogenesis**		
PYGM	Cluster of glycogen phosphorylase, muscle form	0.00056	0.08
AGL	glycogen debranching enzyme isoform X1	0.0003	0.06
LDHB	L-lactate dehydrogenase B chain isoform X1	0.001	−0.07
PPP1R3A	protein phosphatase 1 regulatory subunit 3A	0.0014	0.09
	**Miscellaneous**		
PLIN4	perilipin-4 isoform X5	0.001	−0.12
CA3	carbonic anhydrase 3	<0.0001	0.09

## Data Availability

All data are provided in this manuscript.

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
