# Peer review of "Impact of Coenzyme Q10 Supplementation on Skeletal Muscle Respiration, Antioxidants, and the Muscle Proteome in Thoroughbred Horses"

_antioxidants, 2023, doi:10.3390/antiox12020263_

Round 1

Reviewer 1 Report

Henry et al., manuscript investigates the effects of CoQ10 supplementation on horse skeletal muscle function, particularly in what relates to respiration parameters, antioxidant state and proteome alteration. The study's findings are well-presented and demonstrate the influence of CoQ10 supplementation on various mitochondrial respiration and antioxidant enzyme parameters. Indeed, CoQ10 supplementation results in a preferential utilization of complex II over complex I for energy production. Could this be related to an increase in fatty acid utilization? Moreover, no alterations in total GSH were observed. Do authors predict any alteration in the ratio of GSSG/GSH due to CoQ10 supplementation?

Minor points:

-Line 44, CoQ10 transfer electrons, not protons, from complex I or II to complex III. Please correct text accordingly.

Please confirm whether the volumes used to homogenize tissue samples in the material and methods section were in milliliters or microliters. For instance, authors refer to 500ml in line 198.

Line 179, please define CV

Line 194, please remove chapter 2 at the end of the sentence.

Line 276, please use the order of the figure S1.

In Figure 2 I cannot find the # that is referred in the legend.

Line 331, please correct the first sentence of this paragraph.

Line 341, please correct 2-oxogluterate to 2-oxoglutarate

Line 355, in legend of Figure 5 correct 2-oxogluterate

Author Response

Henry et al., manuscript investigates the effects of CoQ10 supplementation on horse skeletal muscle function, particularly in what relates to respiration parameters, antioxidant state and proteome alteration. The study's findings are well-presented and demonstrate the influence of CoQ10 supplementation on various mitochondrial respiration and antioxidant enzyme parameters. Indeed, CoQ10 supplementation results in a preferential utilization of complex II over complex I for energy production.

Could this be related to an increase in fatty acid utilization?

Added line 441

Moreover, no alterations in total GSH were observed. Do authors predict any alteration in the ratio of GSSG/GSH due to CoQ10 supplementation?

Added line 462

Minor points:

-Line 44, CoQ10 transfer electrons, not protons, from complex I or II to complex III. Please correct text accordingly.

changed

Please confirm whether the volumes used to homogenize tissue samples in the material and methods section were in milliliters or microliters. For instance, authors refer to 500ml in line 198.

Changed 198, 202, 204,

Line 179, please define CV

added

Line 194, please remove chapter 2 at the end of the sentence.

removed

Line 276, please use the order of the figure S1.

We added changed and it is now Figure 2 not a supplemental figure

In Figure 2 I cannot find the # that is referred in the legend.

For clarity the # was removed from the figure 2F (now 3F) legend and removed from in front of the treatment P value as the treatment effect is overruled by the effect of treatment × period effect

Line 331, please correct the first sentence of this paragraph.

changed

Line 341, please correct 2-oxogluterate to 2-oxoglutarate

changed

Line 355, in legend of Figure 5 correct 2-oxogluterate

changed

Reviewer 2 Report

This is a carefully performed study of effects of Coenzyme Q10 (CoQ10) supplementation on muscle mitochondrial function, antioxidant capacity and proteome. A strength is that the study is performed on horses rather than rodents. Adequate methods are used, methodological details are clearly described, and solid results are presented.

I have two major concerns:

1.       The statistical method (except for the proteomics) seems too complicated, especially with only 6 individuals, and makes the presentation of the results difficult to grasp. The hypotheses presented at the end of the Introduction are totally related to effects of CoQ10 treatment, and not to effects of period or period x treatment. Why not simply use one-way RM ANOVA to assess any differences between CoQ10 treatment and placebo. The effect at the individual level can be shown by connecting the two data points for each horse with a line, and open-closed symbols show the order of treatment. In this way, readers can easily assess the results. The text in the Results section can then be focused on describing important statistically significant (or lack of) effects, and the complicated description of period and period x treatment effects is avoided.

2.       To me, the major and highly important result is that the effect of CoQ10 treatment is very limited, i.e., no difference between CoQ10 supplementation and placebo for the majority of measures and only 10% or less where a difference was detected. These results add to several recent reports showing that the effects of CoQ10 supplementation, as well as treatment with other antioxidants, have little effect and these are not always beneficial. This is important since the general belief, even among many scientists, is that antioxidant supplementation has major health promoting effects. In my opinion, the minor differences between CoQ10 treatment and placebo are overinterpreted in the ms, including some conclusions based on trends rather than statistical significance. Thus, the impact of the paper would be markedly improved if the Discussion and conclusions were focused on the fact that CoQ10 treatment had very limited effect, and the limited differences were only briefly discussed.

Minor points:

1.       Line 32: remove the first CoQ10

2.       Line 63: our rather than the authors’ knowledge. The authors’ can be interpreted as the authors of Ref 4.

3.       Line 73: I cannot find any plasma measurements

4.       Lines 109-110: Difficult to understand “day 30 and 120 prior to daily CoQ10 supplementation”. Were not the samples taken at the end of the first and second treatment periods, i.e., no supplementations were given on the days of sampling?

5.       Line 131: “data were”

6.       Lines 142-143: move (E) to after capacities

7.       Headings 2.6 and 2.9: All measurements were done on muscle and hence “Muscle” at the start of these headings is not needed.

8.       Lines 249-250: Spectrum data …

9.       Line 259: use p or P throughout; “data are”

10.   Figure 1, y-axis: Concentrations are about 10x larger than in your previous study (Ref 3) – please check. It would be easier for the reader if the axis unit was “ug/mg protein”

11.   Line 274: HRR is missing in the title

12.   Line 275-277: No reason for presenting CS and CCO activities as supplementary data – awkward to have to change file to assess these important results.

13.   Line 282: P<0.08 does not really make sense. Either it is =0.08 or <0.08. Same concern also later in ms.

14.   Lines 284 and 292: P=0.05 would mean that it did not reach statistical significance. i.e., p<0.05.

15.   Line 331: Change to “In total, 834 unique proteins were identified. Of these, “

16.   The presentation of the proteomics data would be easier to grasp if down- and upregulated proteins were clearly separated instead of “Other differentially expressed” (line 336)

Author Response

I have two major concerns:

  1. The statistical method (except for the proteomics) seems too complicated, especially with only 6 individuals, and makes the presentation of the results difficult to grasp. The hypotheses presented at the end of the Introduction are totally related to effects of CoQ10 treatment, and not to effects of period or period x treatment. Why not simply use one-way RM ANOVA to assess any differences between CoQ10 treatment and placebo. The effect at the individual level can be shown by connecting the two data points for each horse with a line, and open-closed symbols show the order of treatment. In this way, readers can easily assess the results. The text in the Results section can then be focused on describing important statistically significant (or lack of) effects, and the complicated description of period and period x treatment effects is avoided.

The suggested approach would be easier, however, we often see a training effect in horses with antioxidant supplementation. Although horses were fit at the beginning of the trial, Period 2 represents horses being in training longer than in Period 1. Because of this we felt that a period effect might be relevant. We left the statistics as analyzed because it is possible to see impacts of supplementation in a less-trained state, which disappear after training adaptations occur.

  1. To me, the major and highly important result is that the effect of CoQ10 treatment is very limited, i.e., no difference between CoQ10 supplementation and placebo for the majority of measures and only 10% or less where a difference was detected. These results add to several recent reports showing that the effects of CoQ10 supplementation, as well as treatment with other antioxidants, have little effect and these are not always beneficial. This is important since the general belief, even among many scientists, is that antioxidant supplementation has major health promoting effects. In my opinion, the minor differences between CoQ10 treatment and placebo are overinterpreted in the ms, including some conclusions based on trends rather than statistical significance. Thus, the impact of the paper would be markedly improved if the Discussion and conclusions were focused on the fact that CoQ10 treatment had very limited effect, and the limited differences were only briefly discussed.

A paragraph to this effect has been added at the end of the discussion.

Minor points:

  1. Line 32: remove the first CoQ10

removed

  1. Line 63: our rather than the authors’ knowledge. The authors’ can be interpreted as the authors of Ref 4.

changed

  1. Line 73: I cannot find any plasma measurements

removed

  1. Lines 109-110: Difficult to understand “day 30 and 120 prior to daily CoQ10 supplementation”. Were not the samples taken at the end of the first and second treatment periods, i.e., no supplementations were given on the days of sampling?

We have reworded this. Thank you CoQ10 was given on the day of sampling but we wanted to assess levels prior to supplementation.

  1. Line 131: “data were”

changed

  1. Lines 142-143: move (E) to after capacities

changed

  1. Headings 2.6 and 2.9: All measurements were done on muscle and hence “Muscle” at the start of these headings is not needed.

changed

  1. Lines 249-250: Spectrum data …

changed

  1. Line 259: use p or P throughout; “data are”

Changed throughout

  1. Figure 1, y-axis: Concentrations are about 10x larger than in your previous study (Ref 3) – please check. It would be easier for the reader if the axis unit was “ug/mg protein”

We checked the concentrations and they were correct and Changed the axis as suggested. The horses in our previous study were in more intense training which could have been a factor and there is also considerable interindividual variation in CoQ10 concentrations which could play a role. 

  1. Line 274: HRR is missing in the title

added

  1. Line 275-277: No reason for presenting CS and CCO activities as supplementary data – awkward to have to change file to assess these important results.

Added to manuscript

  1. Line 282: P<0.08 does not really make sense. Either it is =0.08 or <0.08. Same concern also later in ms.

Lines 290, 317, and 318 were changed to include exact P values for each measure.

  1. Lines 284 and 292: P=0.05 would mean that it did not reach statistical significance. i.e., p<0.05.

Change to P< 0.05 throughout

  1. Line 331: Change to “In total, 834 unique proteins were identified. Of these, “

changed

  1. The presentation of the proteomics data would be easier to grasp if down- and upregulated proteins were clearly separated instead of “Other differentially expressed” (line 336)

changed